# Generating Bone Marrow Chimeric Mouse Using GPR120 Deficient Mouse for the Study of DHA Inhibitory Effect on Osteoclast Formation and Bone Resorption

**DOI:** 10.3390/ijms242317000

**Published:** 2023-11-30

**Authors:** Jinghan Ma, Hideki Kitaura, Fumitoshi Ohori, Takahiro Noguchi, Aseel Marahleh, Ria Kinjo, Kayoko Kanou, Jiayi Ren, Mariko Miura, Kohei Narita, Itaru Mizoguchi

**Affiliations:** 1Division of Orthodontics and Dentofacial Orthopedics, Tohoku University Graduate School of Dentistry, 4-1, Seiryo-machi, Aoba-ku, Sendai 980-8575, Japan; ma.jinghan.s1@dc.tohoku.ac.jp (J.M.); fumitoshi.ohori.t3@dc.tohoku.ac.jp (F.O.); takahiro.noguchi.r4@dc.tohoku.ac.jp (T.N.); ria.kinjou.p5@dc.tohoku.ac.jp (R.K.); kanou.kayoko.s7@dc.tohoku.ac.jp (K.K.); ren.jiayi.p7@dc.tohoku.ac.jp (J.R.); mariko.miura.b5@tohoku.ac.jp (M.M.); kohei.narita.a2@tohoku.ac.jp (K.N.); mizo@tohoku.ac.jp (I.M.); 2Frontier Research Institute for Interdisciplinary Sciences, Tohoku University, Sendai 980-8575, Japan; marahleh.aseel.mahmoud.t6@dc.tohoku.ac.jp

**Keywords:** DHA, chimeric mouse, GPR120, osteoclastogenesis, TNF-α

## Abstract

Docosahexaenoic acid (DHA) is an omega-3 fatty acid that exerts physiological effects via G protein-coupled receptor 120 (GPR120). In our previous studies, we figured out the inhibitory effects of DHA on TNF-α (Tumor necrosis factor-α)-induced osteoclastogenesis via GPR120 in vivo. Moreover, DHA directly suppressed RANKL expression in osteoblasts via GPR120 in vitro. In this study, we generated bone marrow chimeric mice using GPR120 deficient mice (GPR120-KO) to study the inhibitory effects of DHA on bone resorption and osteoclast formation. Bone marrow cells of wild-type (WT) or GPR120-KO mice were transplanted into irradiated recipient mice, which were WT or GPR120 deficient mice. The resulting chimeric mice contained stromal cells from the recipient and bone marrow cells, including osteoclast precursors, from the donor. These chimeric mice were used to perform a series of histological and microfocus computed tomography (micro-CT) analyses after TNF-α injection for induction of osteoclast formation with or without DHA. Osteoclast number and bone resorption were found to be significantly increased in chimeric mice, which did not express GPR120 in stromal cells, compared to chimeric mice, which expressed GPR120 in stromal cells. DHA was also found to suppress specific signaling pathways. We summarized that DHA suppressed TNF-α-induced stromal-dependent osteoclast formation and bone resorption via GPR120.

## 1. Introduction

Docosahexaenoic acid (DHA) metabolism and utilization play important physiological roles in the human body. This is primarily reflected in brain development and function [1], inflammation and immune regulation [2], cardiovascular health [3], and visual development [4,5]. DHA may also prevent neurological and vascular diseases caused by ischemic stroke and prevent dementia [6]. It is important to note the body cannot synthesize DHA on its own. Therefore, we generally need to obtain sufficient DHA by consuming foods rich in Omega-3 fatty acids, such as seaweed, fish, flaxseed, and sunflower seeds [7]. Maintaining a diet and nutritional balance is important to obtain sufficient DHA [8].

G protein-coupled receptor 120 (GPR120), also known as free fatty acid receptor 4, is a G protein-coupled receptor (GPCR) that is expressed primarily in immune cells, the gastrointestinal tract, and adipose tissue [9]. GPR120 is a functional DHA receptor that regulates various physiological processes [10,11]. It plays a vital role in clinical medicine because of its abundance, broad expression, and important molecular and regulatory activities. Hydrophilic medicines or hormones are easily accessible to GPCRs because of their expression on membranes, and the heterogeneity of their expression in different tissues and cell types, which allows for selective targeting [12]. Recently, there has been an increase in research on GPCR, and our previous study concluded DHA prevents bone resorption and osteoclast formation caused by TNF-α via GPR120 [13]. 

Osteoclasts originate from hematopoietic precursor cells and progress through a series of differentiation processes that occur in the bone marrow microenvironment to develop into active bone-resorbing osteoclasts [14]. Macrophage colony-stimulating factor (M-CSF) and receptor activator of nuclear factor (NF)-κB ligand (RANKL) are the two most critical factors that influence osteoclast differentiation and are both produced by mesenchymal cells in the bone marrow microenvironment [15]. M-CSF enhances osteoclast differentiation by boosting osteoclast precursor cell survival and development and activating RANK on hematopoietic cells so that they can respond to RANKL and regulate cytoskeletal modifications associated with bone resorption [16,17]. Tumor necrosis factor-α (TNF-α), a significant cytokine in inflammatory osteolysis [18], is mainly produced by macrophages. Abnormalities in the production of TNF-α are known to cause a variety of diseases such as major depressive disorder [19], cancer [20], Alzheimer’s disease [21], and irritable bowel disease [22]. In vitro and in vivo studies have shown TNF-α may indirectly increase bone resorption by activating mature osteoclasts, promoting the proliferation and differentiation of osteoclast precursors, or exhibiting a substantial effect on osteoblasts [23,24].

Osteoblasts play a crucial role in bone formation, and while they do not constitute a large percentage of total bone cells, they play a significant role in ensuring proper bone development and balance [25,26]. Bone modeling requires constant deposition and resorption on an appropriate surface in a coordinated and highly regulated manner to sustain dynamic homeostasis [27,28]. This process involves two types of cells: osteoblasts, which are primarily responsible for bone creation, and osteoclasts, which are responsible for bone resorption. Each of these processes is rigorously regulated by environmental, hormonal, and other factors [29]. Osteoblasts also play a role in the control of bone resorption by RANKL, which binds to RANK on the surface of pre-osteoclasts, assisting in their differentiation and fusion. In contrast, osteoblasts also release a soluble decoy receptor (osteoprotegerin, OPG) which binds to RANKL and therefore impacts osteoclast differentiation and activation [30].

TNF-α plays a pivotal role in several forms of inflammatory osteolysis. In vitro studies have indicated stromal cells and bone marrow macrophages were involved in TNF-α-induced osteoclastogenesis. To investigate the relative contributions of each cell type to TNF-α-induced osteoclastogenesis in vivo, we generated four kinds of bone marrow chimeric mice by bone marrow transplantation using wild-type (WT) mice and TNF receptors 1, 2 deficient mice [18,31]. Bone marrow cells of WT or TNF receptor 1, 2 deficient donor mice were transplanted into lethal doses of irradiated recipient mice, which were either WT or TNF receptor 1 and 2 deficient mice respectively. As a result, we obtained four types of mice: those in which both macrophages and stromal cells were responsive to TNF-α, in which only macrophages were responsive to TNF-α, in which only stromal cells were responsive to TNF-α, and in which neither macrophages nor stromal cells were responsive to TNF-α. Using these mice, we found TNF-α-responsive stromal cells played an important role in TNF-α-induced osteoclast formation in vivo [18,31]. Thus, the use of chimeric mice obtained by bone marrow transplantation makes it possible to elucidate the relative contributions of substances, such as cytokines, to bone marrow macrophages and stromal cells in vivo.

In a previous study, we established GPR120-KO mice, which were GPR120 encoding gene Ffar4 deficient mice (FFAR4-KO mice) as the research subject [32]. Lipopolysaccharide (LPS) is a major constituent of the cell wall of Gram-negative bacteria and induces inflammation and pathological bone destruction [33]. LPS also induces the production of pro-inflammatory cytokines, such as TNF-α from macrophages and other cells at sites of inflammation. Furthermore, LPS stimulates osteoblasts to produce and express osteoclast-related cytokine RANKL [34]. These cytokines have been linked to LPS-induced osteoclast formation and bone resorption in both in vivo and in vitro studies. We found DHA inhibits LPS-induced osteoclastogenesis and bone resorption in vivo via GPR120. DHA directly suppresses osteoclastogenesis and LPS-induced TNF-α production by macrophages [32]. Furthermore, DHA was found to prevent TNF-α-induced osteoclast formation and bone loss via GPR120 [13]. In this study, we analyze developed bone marrow chimeric mice to assess how DHA eliminates TNF-α-induced osteoclast development and bone degradation in vivo via GPR120. The purpose of this study is to establish whether bone marrow macrophages and stromal cells are DHA targets by investigating their participation in TNF-α-induced osteoclast development in vivo. The recipient mice were exposed to a lethal amount of radiation, and their hematopoietic cells, including macrophages, were eliminated, while their stromal cells remained. After donor mouse bone marrow cells were transplanted into irradiated recipient mice, the resulting chimeric mice contained both recipient-derived stromal cells and donor-derived macrophages. In this study, we generated chimeric bone marrow mouse using GPR120 deficient mice to study the inhibitory effects of DHA on TNF-α-induced osteoclast formation and bone resorption. However, the GPR120-mediated mechanism by which DHA interferes with RANKL expression in osteoblasts remains unclear. We isolated osteoblasts to demonstrate how DHA affected the activity of the NF-kB and mitogen-activated protein (MAPK) pathways.

## 2. Results

### 2.1. Transplantation of Bone Marrow Cells Related to Different Types of Bone Marrow Chimeric

To investigate the mechanism by which DHA inhibits osteoclast formation via GPR120, four chimeric mice were bred as shown in the flow diagram. Bone marrow cells were extracted from the tibia and fibula of WT and GPR120-KO donor mice. These cells were then transplanted into radiation-exposed recipient mice via caudal intravenous injection. Thus, four different types of chimeric mice were generated by transplanting WT bone marrow cells into irradiated WT mice (WT>WT), GPR120-KO bone marrow cells into irradiated WT mice (KO>WT), WT bone marrow cells into irradiated GPR120-KO mice (WT>KO), and GPR120-KO bone marrow cells into irradiated GPR120-KO mice (KO>KO) (Figure 1A). Primers used to detect GPR120 expression are shown in Figure 1B. WT>WT and WT>KO, but not KO>WT or KO>KO, were detected using PCR (Figure 1C). PCR results demonstrated both WT and GPR120-KO bone marrow macrophages were successfully transplanted into recipient mice. In other words, the WT>WT and WT>KO groups had WT bone marrow macrophages derived from bone marrow cells that could detect GPR120, but the bone marrow macrophages derived from bone marrow cells of the KO>WT and KO>KO groups bore the GPR120 defect. 

### 2.2. DHA Suppresses Stromal-Dependent TNF-α-Induced Osteoclast Formation via GPR120 Activation in the Chimeric Mouse Model

Only TNF-α was injected into the calvariae of the four types of chimeric mice for five consecutive days. On the sixth day, the calvariae were obtained for histological analysis. Large numbers of multinucleated tartrate-resistant acid phosphatase (TRAP)-positive cells were detected in histological sections of the sutured mesenchyme. Under the same conditions, the results differed when TNF-α and DHA were co-injected. There was a significant increase in the number of TPAP-positive cells in the WT>KO and KO>KO groups. The KO>WT group also showed a slight increase, but not as pronounced as the WT>KO and KO>KO groups (Figure 2A,B). Furthermore, the levels of TRAP, RANKL, and OPG messenger ribonucleic acid (mRNA) expression, and RANKL/OPG extracted from the bones of these injected mice were determined by real-time RT-PCR analysis (Figure 2C). The four groups of chimeric mice injected only with TNF-α displayed no obvious differences in either TRAP, RANKL, OPG, or RANKL/OPG (Figure 2C). However, the mRNA expression of TRAP, RANKL, and RANKL/OPG was lower in the WT>WT and KO>WT groups than those in the WT>KO and KO>KO groups when they received co-injections of TNF-α and DHA. The results of these investigations have shown osteoclast formation induced by TNF-α was restrained after co-administrated with DHA in mice that owned WT stromal cells to a certain extent.

### 2.3. DHA Reduces Stromal-Dependent TNF-α-Induced Bone Resorption Mediated by Osteoclast via GPR120 Activation in the Chimeric Mouse Model 

We used microfocus computed tomography (micro-CT) to scan the calvariae of the mice. To quantify the calvarial damage, we calculated a rectangular area of 50 × 70 pixels centered on the frontal suture. We then calculated the ratio of bone resorption to the total area to determine the rate of bone resorption. TNF-α (100 μg/day) injection resulted in a significant increase in the ratio of bone resorption in the four groups of chimeric mice (Figure 3A). In contrast, the chimeric mice that received both TNF-α (100 μg/day) and DHA (100 μg/day) showed different levels of bone resorption. The rate of bone resorption was significantly higher in the groups of mice in which the receptor was GPR120-KO (Figure 3B). The resulting micro-CT analysis indicated bone resorption was restricted by DHA in a stromal-dependent manner.

### 2.4. DHA Suppressed TNF-α Triggered Osteoblast Related RANKL Expression in WT Osteoblasts 

The expression level of RANKL protein in WT osteoblast was detected by ELISA. It was obvious that being treated with TNF-α led to a significant increase in RANKL protein level compared with the PBS or DHA treated group. However, this increasing trend was blocked by DHA, RANKL protein level showed a markedly decrease when DHA co-administrated with TNF-α together (Figure 4). These results may indicate DHA may be able to suppress RANKL expressed by osteoblast which is stimulated through TNF-α.

### 2.5. DHA Inhibits TNF-α-Induced Phosphorylation of Extracellular Signal-Regulated Kinases (ERK) and c-Jun N-Terminal Kinases (JNK) in Osteoblasts 

Calvariae from newborn WT mice were used to obtain primary osteoblasts. TNF-α (100 ng/mL) with DHA (100 ng/mL) or TNF-α (100 ng/mL) alone was added to the same number of osteoblasts at 0, 5, 15, and 30 min. It was shown phosphorylation of p38 MAPKs, ERK1/2, JNK were rapidly increased in TNF-α activated osteoblasts and the peaks were at 5 or 15 min. However, DHA inhibited the phosphorylation of ERK1/2 and JNK, but not that of p38 (Figure 5A). We also stimulated similar osteoblasts with similar concentrations of TNF-α or TNF-α and DHA. The phosphorylation of IkB showed an obvious increase, with a peak at 5 min. However, the phosphorylation of IkB was slightly decreased by DHA co-administration (Figure 5B). Based on these results we can assume DHA was functional in the phosphorylation process of ERK and JNK.

## 3. Discussion

In recent years, the diverse benefits of DHA for humans and other organisms have been extensively studied [35,36,37]. Our previous investigation showed DHA inhibited TNF-α-induced osteoclast formation and bone destruction in vivo and in vitro via GPR120 [13]. Furthermore, we investigated the target cells of DHA for osteoclast formation under stimulation in vivo. We carried out this chimeric mouse model by creating four types of chimeric mice to look into the function of DHA in the stromal-dependent osteoclast-mediated destruction of bones via GPR120. TNF-α-induced osteoclast recruitment is considered to be the centerpiece of the pathogenesis of inflammation-related diseases. In addition, TNF-α also causes osteoclast-induced bone destruction and inhibits osteoblast differentiation and apoptosis. 

As described above, bone marrow cells divide into osteoclasts and exert a bone destruction effect under the enhancement of both RANKL and TNF-α [38]. We have already drawn conclusions DHA inhibits TNF-α enhanced osteoclast formation through GPR120, which included a decrease of the RANKL level produced by osteoblast and the stimulation effect of RANKL on osteoclast formation directly (Figure 6). In this study, we continue to discuss how this mechanism is accomplished. So, we stimulated chimeric mice with TNF-α and observed their calvariae. All four types of chimeric mice had comparable levels of osteoclastogenesis and bone resorption. However, when we stimulated the four chimeric mice with TNF-α and DHA at the same time, the four groups of mice produced extremely distinct results. The WT>KO and KO>KO groups showed substantial osteoclastogenesis and obvious bone resorption in the sagittal suture. Conversely, osteoclast formation and bone destruction were significantly diminished in the KO>WT group, and even more so in the WT>WT group. These results indicated TNF-α-induced osteoclast-mediated bone resorption was inhibited in both WT>WT and KO>WT groups with simultaneous injection of TNF-α and DHA, and this inhibition may be more evident in the WT>WT group. The inhibitory impact of DHA, on the other hand, was not reflected in the WT>KO or KO>KO groups. To further substantiate these results, we also compared the number of osteoclast formations and the percentage of bone resorption in the sagittal suture between the same kind of chimeric mice with different reagents injected. It was demonstrated DHA significantly decreased osteoclast formation and bone resorption in WT>WT and KO>WT groups but not in WT>KO or KO>KO groups (Appendix A). From this series of results, we can make the following inferences. At the time of bone marrow cell transplantation, the hematopoietic cells of recipient mice, including bone marrow macrophages, were destroyed by lethal doses of irradiation, whereas stromal cells survived [18]. Irradiated recipient mice were transplanted with bone marrow cells. Accordingly, the obtained chimeric mice contained recipient-derived stromal cells and donor-derived macrophages. Combining the results of the histological analysis and micro-CT images, we can draw the following conclusions: When both TNF-α and DHA stimulation are applied, DHA inhibition of TNF-α-induced osteoclast-mediated bone resorption occurred mainly in the group with WT stromal cells, instead of the group possessing the GPR120-KO stromal cells. Thus, we can speculate stromal cells contribute more to the inhibitory effect of DHA on osteoclast formation and bone resorption than macrophages in vivo.

Although the data from these experiments support the above conclusions, the role of DHA in macrophages and stromal cells requires further investigation. Several studies have shown one of DHA receptors GPR120 was expressed in macrophages [39] and was involved in the regulation of multiple physiological processes and pathological conditions [40]. Stromal cells, including bone marrow stromal cells, adipocytes, and vascular endothelial cells, are multifunctional cell types found in various tissues and organs [41]. DHA reduces the inflammatory response of stromal cells and limits the production of inflammatory cytokines [42]. However, there are few relevant studies in vivo. Therefore, in this study, we modified the macrophage type of chimeric mice by transplantation to compare the effects of macrophages and stromal cells. Both macrophages and stromal cells are targets of DHA and contribute to the inhibition of osteoclast-mediated bone resorption. The results of this study suggest stromal cells contribute more to osteoclast formation than macrophages.

Previous research has reported TNF-α enhanced RANKL expression in osteoblasts, stromal cells [43], osteocytes, [13] and lymphocytes [44]. TNF-α is a strong activator of phosphorylation of p38, ERK, and JNK MAPK, and these are associated with TNF-α function as TNF-α signaling [45]. Moreover, NF-κB, a transcription factor, has been shown to be necessary for TNF-α-induced RANKL upregulation in osteoblasts [46]. In our previous study, DHA was found to suppress RANKL expression in osteoblasts via GPR120 in vitro [13]. We have concluded that DHA suppressed TNF-α-triggered osteoblast related RANKL expression in WT osteoblasts (Figure 4). To prove DHA-GPR120 axis regulates TNF-α-induced RANKL expression in osteoblasts, we examined the expression levels of RANKL protein in GPR120-KO mice osteoblasts under the same conditions by ELISA. These results showed that RANKL expression was at the same level in TNF-α and TNF-α+DHA groups in GRR120-KO osteoblast, which concluded that DHA does not suppress TNF-α triggered osteoblast related RANKL expression in GPR120-KO osteoblasts (Appendix A). The results suggested that DHA-GPR120 axis regulates RANKL expression in osteoblasts. Therefore, in the current work, we examined the effects of DHA on p38, JNK, and ERK MAPK, and IκB activation by TNF-α. WT group which showed an obvious reduction in phosphorylation after adding DHA. These results indicate that DHA inhibits TNF-α-induced ERK, JNK, and IκB phosphorylation in osteoblasts (Figure 5). We also used GPR120-KO osteoblast for the same experiment. However, the results of the western blot demonstrated similar TNF-α-induced phosphorylation of p38, JNK, and ERK MAPK, and IκB with or without DHA in GPR120-KO osteoblasts. The results suggested that the DHA-GPR120 axis regulates TNF-α-induced signalings (Appendix A). These results suggested inhibition of TNF-α-induced RANKL expression via inhibition of ERK, JNK, and IκB phosphorylation by DHA in osteoblast may inhibit osteoclast formation in vivo. 

## 4. Materials and Methods

### 4.1. Mice

WT mice were 8–10-week-old male C57BL6/J mice purchased from CLEA Japan (Tokyo, Japan). We generated C57BL6 background GPR120-deficient mice as previously described [32]. All animal care and experiments were approved by the Tohoku University of Science Animal Care and Use Committee. Sigma-Aldrich (St. Louis, MO, USA) provided the DHA.

### 4.2. Preparation of Donor Bone Marrow Cells and Creation of Chimeric Mice

One WT and one GPR120-KO animal were selected as donor mice for different groups. After the mice were euthanized, their skin was disinfected with 70% alcohol before the skin and muscles were gently separated using sterile scissors and forceps to expose the tibia and fibula, respectively. After the long bone was completely removed, the tip of the mouse leg bone was cut with sterile scissors to reveal the bone marrow cavity. After flushing with sterile phosphate-buffered saline (PBS), the acquired bone marrow cells were collected into a 1.5 mL centrifuge tube for injection. The recipient mice were also prepared simultaneously. Both WT and GPR120-KO mice were randomly divided into two groups, resulting in a total of four groups. A specific dose of gamma radiation (10 Gy) was administered to each mouse in each of the four groups. Immediately afterward, bone marrow cells from donor WT or GPR120-KO mice were transplanted into each of the recipient mice that were exposed to radiation by tail vein injection (per 100 µL PBS contained 1×10^6^ cells) [18]. We obtained four distinct chimeric mice: bone marrow cell transplantation from WT into WT mice, bone marrow cell transplantation from GPR120-KO into WT mice, bone marrow cell transplantation from WT into GPR120-KO mice, and bone marrow cell transplantation from GPR120-KO into GPR120-KO mice.

### 4.3. Histological Analysis

The four groups of chimeric mice described above were subjected to a series of in vitro experiments. TNF-α (3 µg/day), DHA (100 µg/day) + TNF-α (3 µg/day) together were injected supracalvarial into the four groups of chimeric mice daily for five days. Following th at, the mice were euthanized, and their calvariae were removed and fixed for 24 h at 4 °C in 4% PBS-buffered formaldehyde. After washing the skulls three times, they were demineralized for three days on a shaker at room temperature with 14% ethylenediaminetetraacetic acid (EDTA). After 32 h of dehydration using a tissue processor, the calvariae were separately immersed in paraffin and cut into pieces perpendicular to the sagittal suture (5 µm thickness) using a slicer (TP1020; Leica, Wetzlar, Germany). Following the above processes, paraffin sections were stained using a TRAP Stain Kit (Wako, Osaka, Japan), and all sections were stained with hematoxylin. Osteoblasts are TRAP-positive cells with three or more nuclei. Under an electron microscope, TRAP-positive osteoclasts in the sagittal suture mesenchyme of the mice were measured and standardized to a certain surface area for the TRAP-positive cell count [47].

### 4.4. Micro-CT Analysis of Bone Destruction Area

Calvariae of chimeric mice that only received TNF-α injections or co-administered DHA and TNF-α were removed and scanned using micro-CT (ScanXmate-E090; Kansekan, Yokohama, Japan). To observe bone resorption after supracalvarial injection, three-dimensional images of the calvariae were drawn using the TRI/3DBON64 software (TRI/3D-BON R.7.00.06.0-H-64, RATOC Systems Engineering, Tokyo, Japan). Lastly, the percentage of bone resorption on the surface of the calvariae relative to the total area was calculated using ImageJ (NIH, Bethesda, MD, USA) [47].

### 4.5. RNA Isolation and Real-Time RT-PCR Analysis

Chimeric mice calvariae which were exposed to TNF-α or TNF-α plus DHA for five days were removed and frozen at −80 °C. The cell disrupter (Micro Smash MS-100R; Tomy Seiko, Tokyo, Japan) smashed the bone tissue in TRIzol reagent (Invitrogen, Carlsbad, CA, USA) for RNA extraction. The RNeasy Mini Kit (Qiagen, Hilden, Germany) was used to extract RNA from crushed tissue. The complementary deoxyribonucleic acid (cDNA) was synthesized using Superscript IV reverse transcriptase with the same amount of total RNA. TNF-α, RANKL, and OPG mRNA levels were measured by real-time RT-PCR on a thermocycler dice real-time system (Takara). The PCR cycling settings were as follows: an initial denaturation step at 95 °C (10 s), followed by 45–60 amplification cycles with a denaturation step at 95 °C (5 s), and an annealing step at 60 °C (30 s). The relative expression levels of RANKL mRNA were determined using the reference gene, glyceraldehyde 3-phosphate dehydrogenase (GAPDH). The following primer sequences were used for cDNA amplification: GAPDH, 5′-GGTGGAGCCAAAAGGGTCA-3′ and 5′-GGGGGCTAAGCAGTTGGT-3′; TRAP, 5′-AACTTGCGACCATTGTTA-3′ and 5′-GGGGACCTTTCGTTGATGT-3′; and RANKL, 5′-CCTGAGGCCAGCCATTT-3′ and 5′-CTTGGCCCAGCCTCGAT-3′; and OPG, 5′ -CTTAGGTCCAACTACAGAGGAAC- 3′and 5′ -ATCAGAGCCTCATCACCTT-3′ [48].

### 4.6. Electrophoretic Analysis of PCR Amplification Products

Bone marrow cells from each chimeric mouse were cultured in M-CSF-containing culture media for three days. The non-adherent cells were removed and the RNA was extracted from the adherent cells. Bone marrow macrophages were extracted RNA and cDNA was prepared using the method described above. We prepared the PCR master mix using the KAPA2G Fast HotStart ReadyMix W/Dye PCR Kit (Roche, Indianapolis, IN, USA). The PCR cycling settings were as below: an initial denaturation step at 95 °C (1 min), followed by 45 amplification cycles with a denaturation step at 95 °C (15 s), and an annealing step at 60 °C (15 s) then 72 °C (15 s). The following primer sequences were used for cDNA amplification: GAPDH, 5′-GGTGGAGCCAAAAGGGTCA-3′ and 5′-GGGGGCTAAGCAGTTGGT-3′; GPR120, 5′-GGCACTGCTGGCTTTCATA-3′ and 5′-GATTTCTCCTATGCGGTTGG-3′. Electrophoresis was performed using a Mupid-exu Submarine electrophoresis system with 6-well-agarose gels in 1X TBE buffer. Depending on the position of the indicator, the electrophoresis was terminated after 20 min. After gel removal, the DNA fragments were exposed to UV light for observation. The DNA fragment appeared as an orange fluorescent band under UV light and as bright bands in the image.

### 4.7. Preparation of Osteoblasts

Calvariae from newborn WT mice (<5 days old) were used to obtain primary osteoblasts. Collagenase solution (0.2% *w*/*v*; Wako Pure Chemical Industries, Osaka, Japan) was prepared in an isolation buffer before use [49]. EDTA was produced in PBS and filtered through a 0.2 μm filter at a concentration of 5 mM and 0.1% bovine serum albumin (BSA). Fractions 1 (collagenase), 2 (EDTA), 3 (collagenase), and 4 (collagenase) were collected and digests were gathered [49]. To obtain a high osteoblast fraction, fractions 3–5 were collected. For one night at 37 °C, these cells were nurtured in fetal bovine serum (FBS) 10%, 100 IU/mL penicillin G, and 100 g/mL streptomycin in alpha minimum essential medium (α-MEM) (Wako). Trypsin-EDTA was used to separate the adherent cells (Life Technologies, Grand Island, NY, USA). α-MEM with 10% FBS was used to cultivate the cells for three days, with the medium changed every other day. Osteoblasts were collected from the adherent cells.

### 4.8. Western Blotting Analysis

The purpose of the western blotting analysis was to see how DHA affected the phosphorylation of p38, ERK1/2, JNK MAPK, and IκBα in osteoblasts. Osteoblasts of the same density were cultured separately in ten 60 mm culture dishes, and TNF-α (100 ng/mL) with or without DHA (100 ng/mL) was added to the culture dishes at specific times. The control wells (0 min) did not contain TNF-α or DHA. Osteoblasts isolated from neonatal mice were starved for 3 h in serum-free media in 60 mm cell culture dishes. Cells were lysed for 15 min using radioimmunoprecipitation (RIPA) assay buffer (Millipore, Burlington, MA, USA) containing 1% protease and phosphatase inhibitors (Thermo Fisher Scientific, Waltham, MA, USA), and the insoluble products were separated immediately afterward by centrifugation. Total protein concentrations were quantified using a Pierce BCA protein assay kit (Thermo Fisher Scientific). Depending on the amount of sample an equal proportion of b-mercaptoethanol (Bio-Rad, Hercules, CA, USA) and Laemmli sample buffer (Bio-Rad) were mixed with protein samples before boiling at 95 °C for 5 min. Equal amounts of protein were loaded into ten-well Mini-PROTEAN TGX precast gels (BioRad, CA, USA), transferred to a PVDF Trans-Blot Turbo Transfer System (Bio-Rad, CA, USA), and then incubated in BlockAce (DS Pharma Biomedical, Osaka, Japan) for 1–2 h at room temperature [49]. Membranes were incubated with the following antibodies: Phospho-p44/42 MAPK (Erk1/2) (Thr202/Tyr204) antibody, p44/42 MAPK (Erk1/2) (137F5) Rabbit mAb, Phospho-p38 MAPK (Thr180/Tyr182) (D3F9) XP rabbit mAb, p38 MAPK (D13E1) XP rabbit mAb, Phospho-SAPK/JNK (Thr183/Tyr185) (98F2) rabbit mAb, SAPK/JNK rabbit antibody, Phospho-IκBα (Ser32) (14D4) Rabbit mAb, and IkBα rabbit Ab (Sigma-Aldrich, MO, USA) at a dilution 1:1000 overnight at 4 °C. The membranes were washed in Tris-buffered saline containing Triton X-100 (TBS-T) and incubated for 1 h at room temperature with horseradish peroxidase-conjugated anti-rabbit antibody (Cell Signaling Technologies, Danvers, MA, USA). The signal was detected and the density was measured using an improved chemiluminescence detection system (Thermo Fisher Scientific).

### 4.9. ELISA Assay

Osteoblasts of WT mice were cultured for 3 days. Supernatants of these culture media were gathered respectively for measuring the concentration of RANKL. Ab100749-RANKL (TNFSF11) Mouse ELISA Kit (Abcam, EU, Cambridge, UK) was used in this study. After preparing the standards, 100 μL of standards were added into each well with samples together and incubated for two and a half hours at room temperature. Then discard the solution and wash it 4 times with 1X Wash solution. Complete the removal of all liquid at each step and add 100 μL of 1X Biotinylated RANKL Detection Antibody. Incubated at room temperature for one hour with shaking. Repeated the wash after discarding the solution. Then add 100 μL of 1X HRP-Streptavidin solution to every well and incubate for another 45 min. Wash again before adding TMB One-step Substrate Reagent and shaking for 30 min. Finally, it was read at 450 nm as soon as 50 μL Stop solution was added. 

### 4.10. Statistical Analysis

All data are expressed as mean ± standard deviation (SD). The statistical significance of differences was determined using Scheffé’s test and Multiple *t* test. Statistical significance was set at *p* < 0.05. 

## 5. Conclusions

Considering all the findings of the study, we can summarize that DHA suppresses TNF-α-induced stromal-dependent osteoclast formation and bone resorption via GPR120. DHA was supposed to restrain osteoblast-related osteoclast formation and bone resorption by dampening the MAPK and NF-kB pathways.

## Figures and Tables

**Figure 1 ijms-24-17000-f001:**
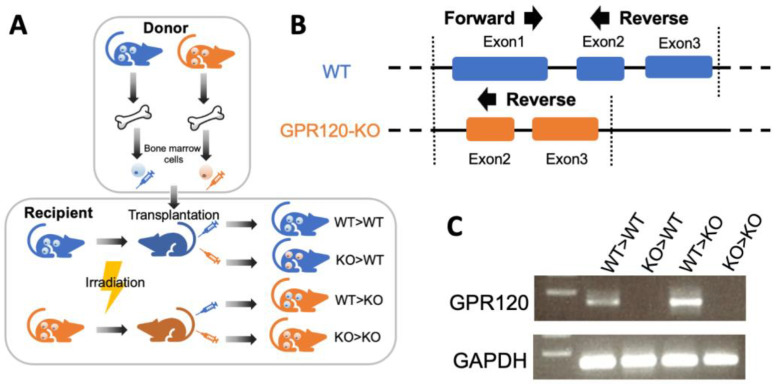
Transplantation of bone marrow cells in different types of bone marrow chimeric mice. (**A**) Diagram of the generation of four types of chimeric mice. (**B**) Primers used for GPR120 gene expression detection. (**C**) The bands of polymerase chain reaction (PCR) product of bone marrow macrophages derived from bone marrow cells of four groups of chimeric mice. The bars are expressed as means and error bars represent SD. The statistical significance was determined using Scheffé’s tests (*n* = 4).

**Figure 2 ijms-24-17000-f002:**
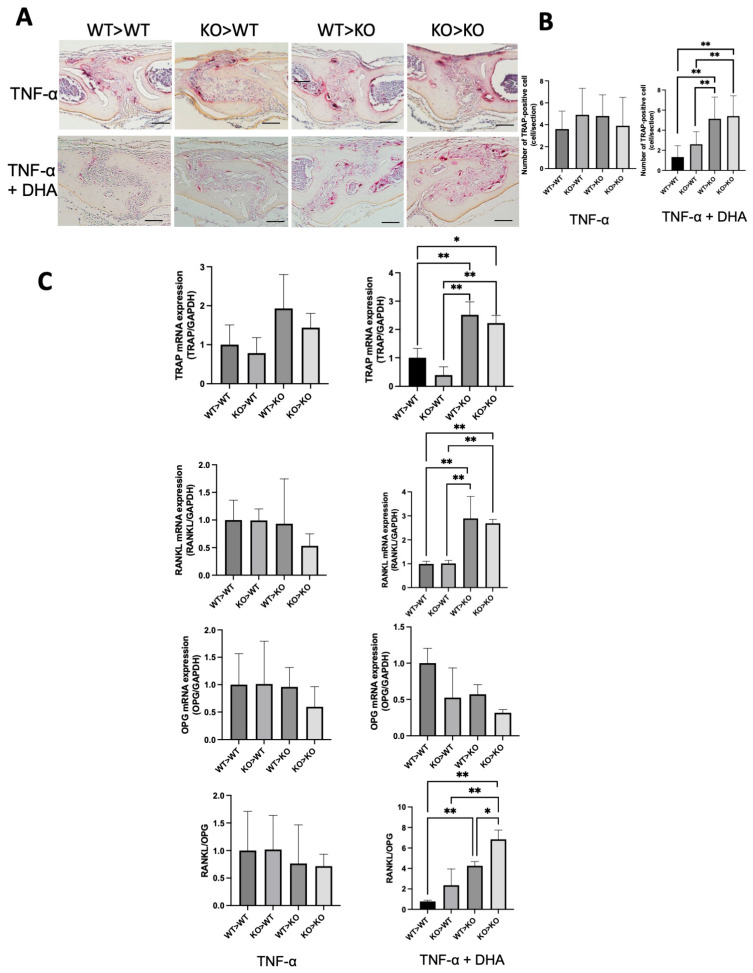
Docosahexaenoic acid (DHA) suppresses stromal-dependent tumor necrosis factor-α (TNF-α)-induced osteoclast formation via GPR120 activation in chimeric mice. (**A**) Photomicrograph of tartrate-resistant acid phosphatase (TRAP) stained sections of mouse calvariae. Calvariae from chimeric mice injected daily with subcutaneous supracalvarial injection of TNF-α only or TNF-α + DHA for five days were sectioned 5-µm-thick. (**B**) Numbers of TRAP-positive cells in the sagittal suture of four groups of chimeric mice administered TNF-α or TNF-α and DHA together. (**C**) TRAP, nuclear factor kappa-Β ligand (RANKL), osteoprotegerin messenger ribonucleic acid (OPG mRNA) levels, and RANKL/OPG in TNF-α only or TNF-α and DHA co-administered chimeric mice chimeric mice calvariae determined by real-time reverse transcription–polymerase chain reaction RT-PCR. The bars are expressed as means and the error bars represent SD. The statistical significance of differences was determined with Scheffé’s tests. (* *p* < 0.05, ** *p* < 0.01; *n* = 4). Scale bars = 50 µm.

**Figure 3 ijms-24-17000-f003:**
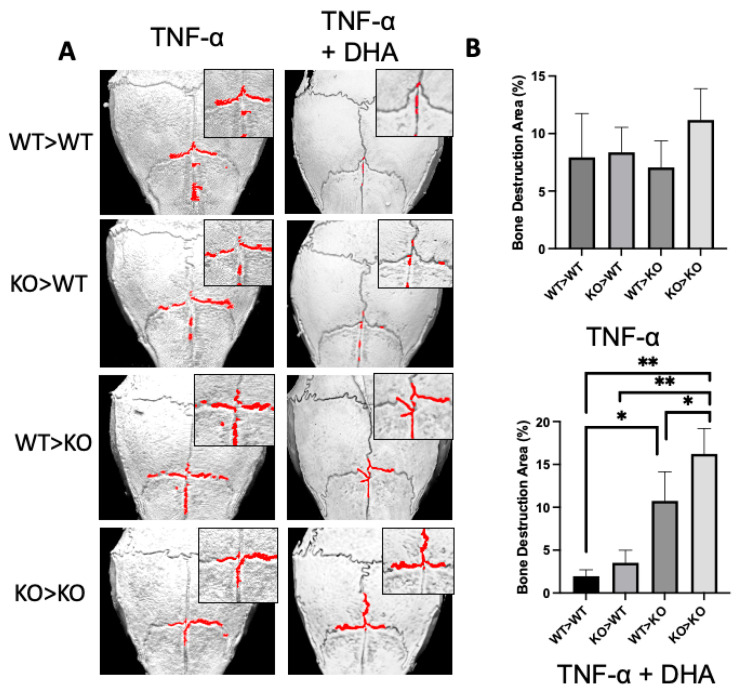
DHA reduces stromal-dependent TNF-α induced bone resorption mediated by osteoclast via GPR120 activation in chimeric mice. (**A**) Three-dimensional reconstructed images of chimeric mouse calvariae. After being injected with TNF-α or TNF-α + DHA for five days, calvariae were resected and scanned by microfocus computed tomography (micro-CT). (**B**) The ratio of bone resorption area to total calvarial bone area. The bars are expressed as means and error bars represent SD. The statistical significance of differences was determined with Scheffé’s tests. (* *p* < 0.05, ** *p* < 0.01; *n* = 4).

**Figure 4 ijms-24-17000-f004:**
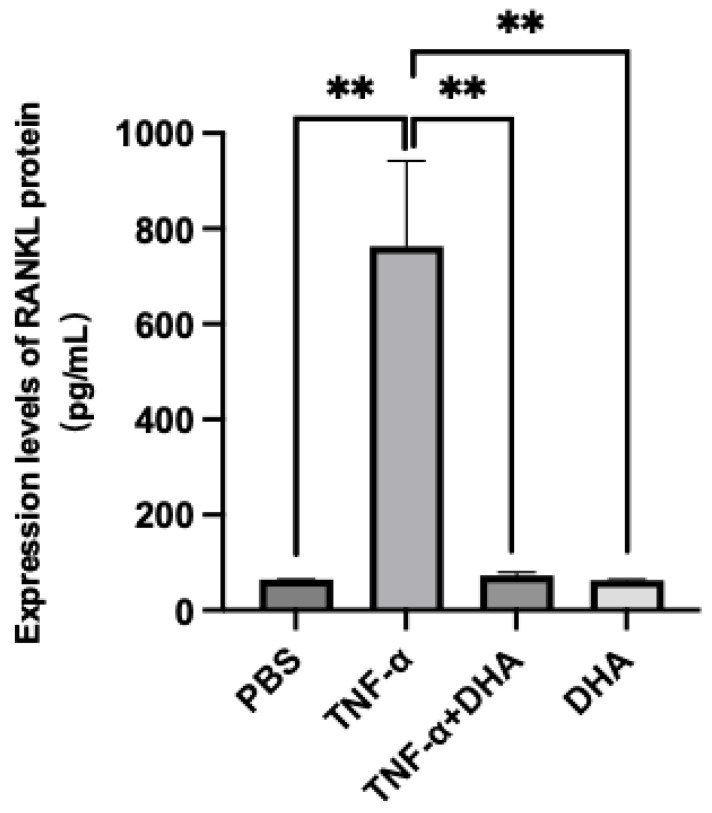
Expression levels of RANKL protein in WT osteoblasts. Administrated with DHA suppressed RANKL level expressed by osteoblast which is triggered by TNF-α. The bars are expressed as means and error bars represent SD. The statistical significance of differences was determined with Scheffé’s tests. (** *p* < 0.01; *n* = 4).

**Figure 5 ijms-24-17000-f005:**
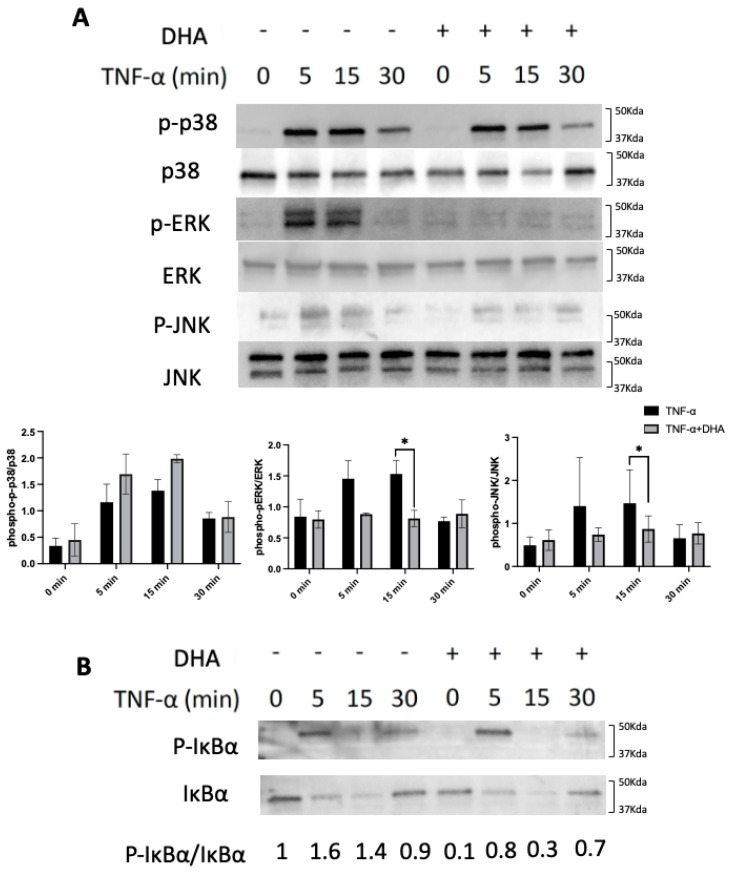
DHA inhibits TNF-α-induced ERK and JNK in osteoblasts. Osteoblasts were incubated with TNF-α (100 ng/mL) with DHA (100 ng/mL) or TNF-α (100 ng/mL) only. Osteoblasts were lysed then analyzed by Western blotting using (**A**) p38, phospho−p38, ERK1/2, phospho−ERK1/2, JNK, phospho−JNK, (**B**) IkB and phospho−IkB antibodies. The bars are expressed as means and error bars represent SD. The statistical significance of differences was determined with Multiple *t* tests. (* *p* < 0.05; *n* = 3).

**Figure 6 ijms-24-17000-f006:**
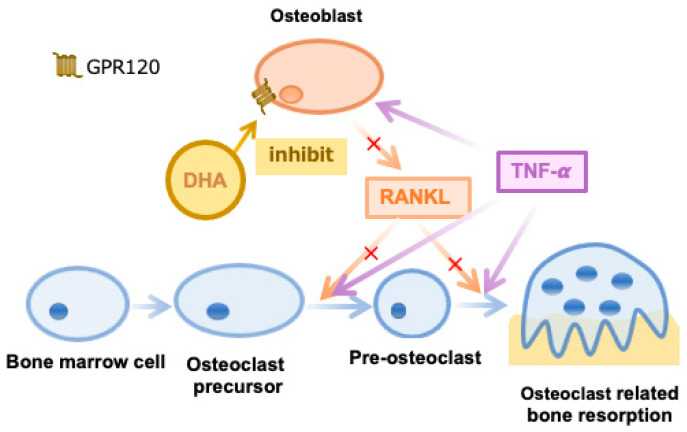
Illustration of the formation of osteoclasts and the role osteoblasts perform in this process. The red crosses imply the reduction of RANKL expression.

## Data Availability

The data presented in this study are available upon request from the corresponding author.

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
