# Peer review of "Generating Bone Marrow Chimeric Mouse Using GPR120 Deficient Mouse for the Study of DHA Inhibitory Effect on Osteoclast Formation and Bone Resorption"

_ijms, 2023, doi:10.3390/ijms242317000_

Round 1
Reviewer 1 Report (Previous Reviewer 2)
Comments and Suggestions for Authors
In this research, Ma and colleagues delved into the mechanism by which DHA influences osteoclast differentiation via GPR120 in living organisms. They also conducted an experiment involving TNF-a-induced bone damage, employing a model that utilized bone marrow chimera mice with donors and recipients categorized as either wild-type or GPR120-deficient. I have some remarks and inquiries regarding this investigation.
In this paper, the authors highlighted that TNF-a-induced osteoclastogenesis and concomitant bone destruction were mitigated by the co-administration of DHA, as depicted in figure 2 and figure 3 (line 165-167, and line 193-194). However, the authors presented the effect of TNF-a with/without DHA in separated graphs, without conducting a statistical comparison between TNF-a single administration and TNF-a + DHA co-administration groups. It is important that the authors refrain from making claims without the support of statistical comparison.
The central question in this study is whether DHA-GPR120 axis regulates osteoclastic bone destruction. Using an in vivo experiential system, the authors got a clue that the DHA-GPR120 axis suppresses stromal-dependent osteoclast formation. Since the authors have GPR120-deficient mice, there is no reason not to utilize the mice to further elucidate the molecular mechanisms presented in figure 4 and figure 5. This effort will help clarify the authors’ claim in this study.
Author Response
Dear reviewer,
Thank you for your constructive comments. I will answer them in order
- In this paper, the authors highlighted that TNF-a-induced osteoclastogenesis and concomitant bone destruction were mitigated by the co-administration of DHA, as depicted in figure 2 and figure 3 (line 165-167, and line 193-194). However, the authors presented the effect of TNF-a with/without DHA in separated graphs, without conducting a statistical comparison between TNF-a single administration and TNF-a + DHA co-administration groups. It is important that the authors refrain from making claims without the support of statistical comparison.
Thank you for your suggestion, I have conducted a statistical comparison between TNF-a single administration and TNF-a + DHA co-administration groups by T-test. The graphs were added in figure S1 and was discussed in line 274-279 which is marked in red.
- The central question in this study is whether DHA-GPR120 axis regulates osteoclastic bone destruction. Using an in vivo experiential system, the authors got a clue that the DHA-GPR120 axis suppresses stromal-dependent osteoclast formation. Since the authors have GPR120-deficient mice, there is no reason not to utilize the mice to further elucidate the molecular mechanisms presented in figure 4 and figure 5. This effort will help clarify the authors’ claim in this study.
Thank you for your comment. Osteoblasts from GPR120-deficient mice were used for ELISA and western blot, these results were added as supplement Figure S2 and was discussed in line 312-318 which is marked in red. Bar graphs of ELISA experiment were corrected (Figure 4), because we made a wrong calculation in Y axis, but results remain unchanged.
Reviewer 2 Report (New Reviewer)
Comments and Suggestions for Authors
Jinghan Ma et al. generated nice bone marrow chimeric mice model using GPR120 KO and WT mice. They showed that TNF-a injection induced more bone resorption in mice with KO stromal cells compared with mice with WT stromal cells. Mechanistically, they demonstrated that TNF-a stimulated RANKL production in WT osteoblasts. TNF-a stimulated p-p38, p-ERK, p-JNK, and p-IkBa signaling and treatment with DHA suppressed p-ERK, p-JNK, p- p-IkBa (but not p-p38) induced by TNF-a. This is a well-written manuscript that clearly demonstrated the role of GPR120 in osteoclastogenesis and bone resorption.
I have one suggestion to the authors:
On page 5 line 177-178, the authors wrote “ TNF-a and DHA co-administered chimeric mice chimeric mice calvariae”. There was an extra “chimeric mice” in the sentence that need to be deleted.
Comments on the Quality of English LanguageMinor English errors that need to be corrected. On page 177-178, there was an extra chimeric mice in the sentence that need to be deleted.
Author Response
Dear reviewer,
Thank you for your constructive comments.
On page 5 line 177-178, the authors wrote “ TNF-a and DHA co-administered chimeric mice chimeric mice calvariae”. There was an extra “chimeric mice” in the sentence that need to be deleted.
Thank you for your kind suggestion. I have corrected the mistake. It was marked in red in line 177.
This manuscript is a resubmission of an earlier submission. The following is a list of the peer review reports and author responses from that submission.
Round 1
Reviewer 1 Report
Comments and Suggestions for Authors
In their manuscript entitled “Generating bone marrow chimeric mouse using GPR120 deficient mouse for the study of DHA inhibitory effect on osteoclast formation and bone resorption”, Jinghan Ma and colleagues present a short study in which they explored the contribution of bone marrow cells to the protective role of G protein-coupled receptor 120 (GPR120), activated with Docosahexaenoic acid (DHA), against osteoclastogenesis and bone loss in the calvaria of mice treated with TNF-alpha.
The aim of the study is to “establish whether bone marrow macrophages and stromal cells are DHA targets by investigating their participation in TNF-alpha-induced osteoclast development in vivo”
For this they use a GPR210-KO (Ffar4-KO) mouse model the authors established previously (Frontiers in Endocrinology, 2019), with which they showed that DHA inhibits lipopolysaccharide (LPS)-induced bone resorption and osteoclast formation via GPR120 activation.
In the present study, the authors used sub-lethal irradiated mice (WT or GPR210-KO) complemented with bone marrow (WT or GPR210-KO) (Figure 1), according to an approach they have published. They injected TNF-alpha in the calvaria of the mice to induce inflammation and examine the effect of the co-injection of DHA. In the calvaria of 4 animals per group, they counted osteoclasts, measured the levels of TRAP osteoclastic marker, RANKL and OPG by QPCR (Figure 2) and they measured bone destruction (Figure 3). Finally, immunoblot (Figure 4) shows the phosphorylation levels of p38, ERK, JNK and IkappaBalpha in response to TNF-alpha with or without DHA, in osteoblasts derived from WT calvaria.
The conclusion from figures 2 and 3 is that TNF-alpha effects are blocked by DHA only when the irradiated mice are WT, whatever the genotype of the bone marrow donor, suggesting that stromal cells contribute to the inhibitory effect of DHA, but not bone marrow cells, which contain bone marrow macrophages that can be osteoclast precursors.
The conclusion for figure 4 is that DHA inhibited ERK and JNK, but not p38, phosphorylation and slightly decreased IkappaBalpha phosphorylation.
General comments:
The abstract contains only one sentence related to the results. It should reflect the content of the article.
In the introduction lines 94-96, the authors state “we found that TNF-alpha-responsive stromal cells play an important role in TNF-alpha-induced osteoclast formation in vivo [18,31–33]”. There are 4 references cited for this single sentence, it is not clear what is shown in each article.
In the result section, it would be easier for the reader to provide a short conclusion/hypothesis in each paragraph after the description of the data. In the present version, the conclusions of the experiments appear in the discussion section.
This study convincingly shows that stromal cells, rather than bone marrow cells, mediate the inhibitory effect of DHA on TNF-alpha-induced bone degradation (Figures 2 and 3). Still, in their previous report, the authors showed a direct effect of DHA, via GPR210, on osteoclast differentiation stimulated by RANKL or by TNF-alpha, independently from the presence of osteoblasts (Frontiers in Endocrinology, 2019). The authors should discuss their present results considering to this previous work.
To be able to compare more easily between TNF-alpha and TNF-alpha + DHA in Figures 2 and 3, graphs should be presented side by side as in 2B; the scale bars should be identical (same size and same range) in both conditions for the same parameter.
The term Ffar4 is the official name of the gene encoding GPR210 and Ffar4-KO is how the mouse model was termed in the original article (Frontiers in Endocrinology, 2019). Thus, Ffar4 should be mentioned at least once in the article.
Scheffé’s test is parametric and requires normal distribution of the data. Was this condition tested? In fact, this is rarely the case with such low number of animals per group (n=4). A non-parametric (rank) Kruskal-Wallis test appears more appropriate to analyze the data.
Regarding signaling, the immunoblots presented in Figure 4 do not sustain the conclusions. A single experiment is not sufficient to conclude on the subtle effects observed and the quantification method is not explained. The poor quality of JNK and IkappaBalpha immunoblots does not allow drawing any conclusion. The rational for examining these pathways is unclear: the discussion mentions p38, ERK, and JNK MAPK are “associated with TNF-alpha function as TNF-alpha signaling”. This is not informative and should be explained in the context of the biological question here. In any case, it is not possible to conclude “These results indicate that DHA inhibits TNF-α-induced ERK, JNK, and IκB phosphorylation in osteoblasts”. Without at least an immunoblot showing the expression of RANKL in this experiment, it is not possible to relate the results to osteoclast differentiation in vivo, contrarily to what is suggested lines 268-271.
The discussion is mainly the conclusion of the experiments presented in the discussion section. It contains very few references. It has to be developed and expose the meaning of the present finding in the context of the literature.
The authors should comment on the fact that they used LPS for their study in 2019 and TNF-alpha in the present study. The similarities/differences between LPS and TNF-alpha link with DHA/GPR210 should be discussed.
The authors should provide general scheme including TNF-alpha, GPR210, DHA, osteoblasts and osteoclasts to model their hypothesis.
The conclusion lines 396-398 is not sustained by the data: “Considering all the findings of the study, we can summarize that DHA suppresses TNF-alpha-induced stromal-dependent osteoclast formation and bone resorption via GPR120 by dampening the MAPK and NF-kB pathways.”
Author Response
To reviewer 1
Thank you for reading carefully and for the very helpful comments. I will answer them in order
- The abstract contains only one sentence related to the results. It should reflect the content of the article.
Thank you for your comments. As you suggested I have included introductions of the results of these experiments, which have been highlighted in red. (However the abstract should be under 200 words, so it’s difficult to include many details in the abstract.)
- In the introduction lines 94-96, the authors state “we found that TNF-alpha-responsive stromal cells play an important role in TNF-alpha-induced osteoclast formation in vivo [18,31–33]”. There are 4 references cited for this single sentence, it is not clear what is shown in each article.
Thank you for your comment. We have rewritten this part and it has been highlighted in red (line 88 and 95). Two similar citations were deleted.
- In the result section, it would be easier for the reader to provide a short conclusion/hypothesis in each paragraph after the description of the data. In the present version, the conclusions of the experiments appear in the discussion section.
Thanks to your comments, brief conclusions/hypotheses about each result were added at the end of each paragraph (in result 2.3.4, highlighted in red). Since each result has a strong correlation, they are also described in detail in the discussion. It is hoped that the short summary for each paragraph will help the reader understand the conclusions corresponding to each of the figures.
- This study convincingly shows that stromal cells, rather than bone marrow cells, mediate the inhibitory effect of DHA on TNF-alpha-induced bone degradation (Figures 2 and 3). Still, in their previous report, the authors showed a direct effect of DHA, via GPR210, on osteoclast differentiation stimulated by RANKL or by TNF-alpha, independently from the presence of osteoblasts (Frontiers in Endocrinology, 2019). The authors should discuss their present results considering to this previous work.
Thank you for your comments. Just as you have mentioned, we already made a conclusion that the direct suppression effect of DHA, via GPR210, on osteoclast differentiation stimulated by RANKL or by TNF-alpha. We draw this conclusion by in vitro experiments despite the effect of osteoblast. In vivo conditions could be much more flexible and complex than in vitro experiments. That's what we designed this mouse model for. The effect of the direct action of DHA may be masked when multiple factors act together in the formation of osteoclasts. The results drawn in this study were based on the chimeric mice model, and they all point out that stromal cells, rather than bone marrow cells, mediate the inhibitory effect of DHA on TNF-alpha-induced bone degradation.
- To be able to compare more easily between TNF-alpha and TNF-alpha + DHA in Figures 2 and 3, graphs should be presented side by side as in 2B; the scale bars should be identical (same size and same range) in both conditions for the same parameter.
Thank you for your comment, I have changed the layout and the scale of Figure 2 and 3.
- The term Ffar4 is the official name of the gene encoding GPR210 and Ffar4-KO is how the mouse model was termed in the original article (Frontiers in Endocrinology, 2019). Thus, Ffar4 should be mentioned at least once in the article.
Thank you for your suggestion, to increase the correlation between studies we have mentioned Ffar4-KO in (line 99-100) and it has been highlighted in red.
- Scheffé’s test is parametric and requires normal distribution of the data. Was this condition tested? In fact, this is rarely the case with such low number of animals per group (n=4). A non-parametric (rank) Kruskal-Wallis test appears more appropriate to analyze the data.
Thank you for your comment. In our studies, we used the Scheffes Test to analyze multiple comparisons among groups. We checked the normality of each group of data by the Shapiro-Wilk test (W test) which requires a random sample of between 3 and 2000 and is often used when 3≤n≤50 (compared with D’Agostino-Pearson test 50≤n≤1000 or Shapiro-Francia test 5≤n≤5000). The result of the Shapiro-Wilk test (by GraphPad Prism) demonstrated that P values were greater than 0.05, which demonstrated that the data was normal. So we followed the Scheffés test. If it is below 0.05, the Kruskal-Wallis test is better.
- Regarding signaling, the immunoblots presented in Figure 4 do not sustain the conclusions. A single experiment is not sufficient to conclude on the subtle effects observed and the quantification method is not explained. The poor quality of JNK and IkappaBalpha immunoblots does not allow drawing any conclusion. The rational for examining these pathways is unclear: the discussion mentions p38, ERK, and JNK MAPK are “associated with TNF-alpha function as TNF-alpha signaling”. This is not informative and should be explained in the context of the biological question here. In any case, it is not possible to conclude “These results indicate that DHA inhibits TNF-α-induced ERK, JNK, and IκB phosphorylation in osteoblasts”. Without at least an immunoblot showing the expression of RANKL in this experiment, it is not possible to relate the results to osteoclast differentiation in vivo, contrarily to what is suggested lines 268-271.
Thank you for your comment, the experiment was repeated 3 or more times, the quality of the image was improved (new Figure 4), and the quantification method was complemented in Methords 4.8 (line 380-383). Besides, the RANKL expression was checked by ELISA and it was added in Figure 4. It was difficult to evaluate the expression of RANKL by western blot because the amount of RANKL bound on the membrane was too small to detect. Instead of this, we evaluate soluble RANKL expression by RANKL ELISA kit (figure 4.).
- The discussion is mainly the conclusion of the experiments presented in the discussion section. It contains very few references. Ithas to be developed and expose the meaning of the present finding in the context of the literature.
Thank you for your suggestion, we added more citations in discussion. The meaning of the present finding was that this mice model will to some extend model the complex mechanisms in the human body to through light on further research in the clinical application of DHA in the future.
- The authors should comment on the fact that they used LPS for their study in 2019 and TNF-alpha in the present study. The similarities/differences between LPS and TNF-alpha link with DHA/GPR210 should be discussed.
Thank you for the suggestion, we discussed the relationship between TNF-a and LPS RANKL in Introduction (Line 101-106), and LPS with DHA/GPR120 in line 106-108.
- The authors should provide general scheme including TNF-alpha, GPR210, DHA, osteoblasts and osteoclasts to model their hypothesis.
The scheme was added as Figure 6 to discuss the interaction of TNF-alpha, GPR210, DHA, osteoblasts, and osteoclasts. And discussion was added in line 258 -263
- The conclusion lines 396-398 is not sustained by the data: “Considering all the findings of the study, we can summarize that DHA suppresses TNF-alpha-induced stromal-dependent osteoclast formation and bone resorption via GPR120 by dampening the MAPK and NF-kB pathways.”
Thank you for your comment. From the results of this study, this conclusion does have a partial presumption, so we've replaced it with a more appropriate expression. (line 454 -457)
Reviewer 2 Report
Comments and Suggestions for Authors
DHA has been shown to have an impact on osteoclast differentiation via the G-protein-coupled receptor 120 (GPR120). In this study, Ma et al. investigated how DHA regulates osteoclast differentiation through GPR120 in vivo and performed a TNF-a-induced bone destruction model using bone marrow chimera mice of wild-type/GPR120-deficient donors and wild-type/GPR120-deficient recipients. The authors revealed that the DHA-mediated inhibitory effect of TNF-a-induced osteoclast differentiation and bone destruction was mitigated when recipient mice lacked GPR120, suggesting that DHA affects stromal cells, which in turn affects osteoclast formation. The authors also examined the effect of DHA on TNF-a-induced activation of MAPK and NF-kB in calvaria-derived osteoblasts and revealed an inhibitory effect of DHA on the activation of p38, JNK, and NF-kB. This is an interesting study. I have a few comments and questions about this study.
In the last figure, the authors examined the impact of TNF-a and DHA on MAPK and NF-kB activation. However, it is not quite clear how the activation of these signaling molecules indeed regulates osteoblast-mediated osteoclast differentiation via GPR120. To clarify this, it is essential to demonstrate whether the signaling activation (p38, JNK, and NF-kB) by TNF-a and DHA is affected in GPR120-deficient osteoblasts. Additionally, the authors should investigate whether the signaling activation (p38, JNK, and NF-kB) indeed induces upregulation of RANKL in wild-type osteoblasts but not in GPR120-deficient osteoblasts. Furthermore, it would have been better if the authors had quantified the western blot images to provide more reliable and quantitative data.
Comments on the Quality of English LanguageModerate editing of English language is required.
Author Response
To reviewer 2
In the last figure, the authors examined the impact of TNF-a and DHA on MAPK and NF-kB activation. However, it is not quite clear how the activation of these signaling molecules indeed regulates osteoblast-mediated osteoclast differentiation via GPR120. To clarify this, it is essential to demonstrate whether the signaling activation (p38, JNK, and NF-kB) by TNF-a and DHA is affected in GPR120-deficient osteoblasts. Additionally, the authors should investigate whether the signaling activation (p38, JNK, and NF-kB) indeed induces upregulation of RANKL in wild-type osteoblasts but not in GPR120-deficient osteoblasts. Furthermore, it would have been better if the authors had quantified the western blot images to provide more reliable and quantitative data.
Thank you very much for your affirmation and suggestion, as you have mentioned, this study would be more complete if “the signaling activation by TNF-a and DHA is affected in GPR120-deficient osteoblasts” was included. However, there are several reasons why this related experiment is difficult to carry out in this paper.
Firstly, we did a western blot and then performed statistics analysis of the results. All the proteins we used for the western blot were isolated from newborn baby mice no more than 5 days. To get a reliable result the western blot should be repeated again and again, which means a large amount of GPR120-KO baby mice were needed. Unlike WT mice which can be bought at any time, mating and getting enough GPR120-KO baby mice was not easy. Neither the period nor the number of babies can be predicted.
Secondly, to make the results more reliable and compare the difference between WT and GPR120-KO mice signaling activation, all the groups of protein should be isolated under the same condition at the same time and added to the same membrane. Each membrane should contain both WT and GPR120-KO, both TNF-a and DHA at different periods of time. The sample size is too large to get satisfactory results.
Besides, the RANKL expression was checked by ELISA and it was added in Figure 4. It was difficult to evaluate the expression of RANKL by western blot because the amount of RANKL bound on the membrane was too small to detect. Instead of this, we evaluate soluble RANKL expression by RANKL ELISA (figure 4.). We have also repeated all the western blot experiments and got better images (Figure 5). All in all, thank you for making such valuable suggestions.
Reviewer 3 Report
Comments and Suggestions for Authors
This study analyzed the effects of DHA on bone metabolism, and it is expected to provide valuable grounds for presenting the possibility of using DHA by analyzing the effects on the function of the GPR120 receptor, in particular.
The research method, experimental design, summary of the presented results, etc. are well done. However, the most important part of this study is bone metabolism, inflammatory process, and the action on DHA and GPR120 receptor, and the experimental method focuses on these as well.
Therefore, I think that this part should be focused in the discussion process. However, the description of the items related to the DHA action according to TNF-a treatment and GPR120 KO treatment is insufficient. It is requested that the discussion of this part should be clearly explained in more detail.
Comments on the Quality of English LanguageMinor editing English language required.
Author Response
To reviewer 3
However, the most important part of this study is bone metabolism, inflammatory process, and the action on DHA and GPR120 receptor, and the experimental method focuses on these as well.
Therefore, I think that this part should be focused in the discussion process. However, the description of the items related to the DHA action according to TNF-a treatment and GPR120 KO treatment is insufficient. It is requested that the discussion of this part should be clearly explained in more detail.
Thank you very much for your affirmation and suggestion. The discussion about items related to the DHA action according to TNF-a treatment and GPR120 KO treatment has been refined in the new version. line258 -263(figure 6) Highlighted in red.